# Investigation of locomotive syndrome improvement by total hip arthroplasty in patients with hip osteoarthritis: A before-after comparative study focusing on 25-question geriatric locomotive function scale

Shigeaki Miyazaki[1]*, Kurumi Tsuruta[2], Saori Yoshinaga[3], Yoshinori Fujii[4], Amy Hombu[5], Taro Funamoto[6], Takero Sakamoto[6], Takuya Tajima[6], Yoshihiro Nakamura[6], Hideki Arakawa[1], Jun Nakatake[1], Etsuo Chosa[7]

1 Rehabilitation Unit, University of Miyazaki Hospital, Miyazaki, Japan, 2 Department of Nursing, Faculty of Human Health Sciences, Shunan University, Shunan, Yamaguchi, Japan, 3 School of Nursing, Faculty of Medicine, University of Miyazaki, Miyazaki, Japan, 4 Department of Mathematics Education, Faculty of Education, University of Miyazaki, Miyazaki, Japan, 5 Center for Language and Cultural Studies, Lecturer, University of Miyazaki, Miyazaki, Japan, 6 Department of Orthopaedic Surgery, Faculty of Medicine, University of Miyazaki, Miyazaki, Japan, 7 Community Medical Center, University of Miyazaki Hospital, Miyazaki, Japan

* 03-5-23@med.miyazaki-u.ac.jp

## Abstract

### Background

The 25-Question Geriatric Locomotive Function Scale (GLFS-25) is one of the tests used to assess the risk of locomotive syndrome (LS). It is a comprehensive tool for measuring LS improvement after total hip arthroplasty (THA) and provides beneficial information for rehabilitation after THA. The primary objective of this study was to clarify LS improvement in patients with hip osteoarthritis (OA) who have undergone unilateral primary THA using GLFS-25. A secondary objective was to identify the impact of THA on each specific GLFS-25 item for optimizing functional recovery.

### Methods

The participants of this study were 273 patients who underwent primary THA for hip OA. LS was evaluated using the GLFS-25, stand-up test, and two-step test before receiving THA and three months after THA.

### Results

Before THA, items rated as "moderately difficult" (score ≥2) in GLFS-25 included pain-related Q3 and Q4, activities of daily living (ADL)-related Q12, Q13, Q15, and Q18, and social function-related Q21 and Q23. At three months after THA, these subjective symptoms showed significant improvement. Further analysis of the

**Data availability statement:** All relevant data are within the paper and its Supporting Information files.

**Funding:** This work was supported by the Japan Society for the Promotion of Science Grants-in-Aid for Scientific Research (Grant No. 24K20438). The funders had no role in study design, data collection and analysis, decision to publish, or preparation of the manuscript. There was no additional external funding received for this study.

**Competing interests:** The authors have declared that no competing interests exist.

relationship between these subjective symptom improvements and LS improvement revealed that all items, except pain-related Q3, were significantly associated with LS improvement.

## Conclusions

Patients experienced not only severe hip pain and physical discomfort but also significant difficulties with activities of daily living (ADL) and social participation before THA. LS improvement after THA was strongly associated with improvements in the subjective symptoms of ADL and social functioning. Based on these findings, rehabilitation strategies that focus on enhancing mobility, improving ADL and social engagement, and optimizing gait function after THA are crucial for further supporting LS recovery.

## Introduction

Locomotive Syndrome (LS) is a concept introduced by the Japanese Orthopedic Association (JOA) in 2007 and is defined as a condition that reflects a decline in mobility due to locomotive organ impairment [1]. LS is influenced by multiple factors, including osteoarthritis, osteoporosis (OA), degenerative spinal disease, spinal canal stenosis, and sarcopenia. As LS progresses, the risk of requiring future long-term care increases [2–4]. Although LS is widely recognized in Japan, its clinical significance on a global scale remains unclear. Similar conditions, such as sarcopenia and frailty, also describe mobility impairments but differ in the defining characteristics. Sarcopenia is characterized by age-related declines in muscle mass and strength [5], whereas frailty is defined as a broader spectrum of physical, cognitive, and social vulnerabilities in older adults [6]. While sarcopenia is more appropriate for assessing muscle mass and frailty for evaluating nutritional status, LS specifically focuses on locomotive organ dysfunction, making it essential for assessing mobility recovery, particularly following surgical interventions such as total hip arthroplasty (THA) [7,8].

THA is a well-established surgical treatment to restore hip function, relieve pain, and improve the limitations in activities of daily living (ADL) caused by hip OA. Previous cohort studies have reported that the improvement rate in total Clinical Decision Limits (CDL) stage 3 was 46.7% at three months after THA, with significant improvements observed in all LS risk tests, including the Stand-up Test, Two-step Test, and the 25-Question Geriatric Locomotive Function Scale (GLFS-25) [9]. The GLFS-25 is a self-reported assessment tool that comprehensively evaluates pain and quality of life (QOL), originally developed as a screening instrument for older adults with locomotive dysfunction [10,11]. Regarding THA's impact on GLFS-25, previous studies have shown that while 90% of patients with severe hip joint disorders required long-term care before THA, this percentage decreased to 30% three months after surgery [12]. However, it remains unclear which specific GLFS-25 items show the most improvement after THA and how they differ between patients who experience improvement and those who do not.

Although evaluating hip joint function is important, determining which functional abilities are regained is also essential. A comprehensive evaluation using GLFS-25 provides valuable insights into LS improvement after THA and can help refine postoperative rehabilitation strategies. The primary objective of this study was to clarify LS improvement in patients with hip OA who have undergone unilateral primary THA using GLFS-25. A secondary objective was to identify the impact of THA on each specific GLFS-25 item for optimizing functional recovery.

## Materials and methods

### Study design

Ethical statement: This before-after comparative study was approved by the Research Ethics Committee of the University of Miyazaki, School of Medicine (Approval No. O-0783). It was carried out in compliance with the Ethical Guidelines for Medical and Biological Research Involving Human Subjects at the Department of Rehabilitation Medicine, University of Miyazaki Affiliated Hospital. The participants of this study were patients who underwent primary THA for hip OA between October 2018 and May 2024. Approval for this study was obtained from the Research Ethics Committee in September 2020. Therefore, data collected before obtaining approval from the Research Ethics Committee were categorized as a retrospective study, while data collected after obtaining approval were categorized as a prospective study. All data used in the retrospective study were fully anonymized prior to access (Date of data access for the purpose of research: September 1, 2023). As it was not feasible to contact participants in the retrospective study and because the information was initially collected for clinical purposes, the opt-out method was applied in accordance with ethical guidelines. Information regarding the conduct of the research including the objectives was disclosed and the research participants were provided an opportunity to refuse inclusion in the research.

### Study execution

1. Patient selection: The participants in this study were patients who underwent unilateral primary THA for hip OA. The target patients agreed to participate in both before THA and three months after THA evaluations. Patients with femoral head necrosis, trauma, rheumatoid arthritis, or infection, and those with incomplete outcome measure data sets were excluded from the study. As a result of careful selection and rigorous screening, 273 patients were included in this study (Fig 1). The breakdown of the total CDL stages was as follows: 0 patients at stage 0, 1 patient at stage 1, 27 patients at stage 2, and 245 patients at stage 3. The cohort in this study included patients with polyarticular disease and those who have undergone arthroplasty in other joints. Patients who underwent simultaneous bilateral THA were not included.

A previous study measured the spatiotemporal gait parameters of patients undergoing THA and analyzed their gait characteristics using principal component analysis. That study examined the impact of THA-induced gait patterns on LS [13]. Although the data in the present study were also used in the previous study, the objectives of the two studies were different.

2. Surgical approach and post-operative rehabilitation: All cases in this study underwent THA using the anterior minimally invasive surgery (AMIS) approach or transgluteal approach performed by experienced orthopedic surgeons at the hospital of the authors' affiliated institution. The postoperative rehabilitation program was conducted by physiotherapists specializing in musculoskeletal disorders, aiming to reduce pain, increase range of motion, restore neuromuscular coordination, improve gait patterns, and enhance ADL. Rehabilitation after THA began the day after THA and continued for three months. Patients received one-on-one rehabilitation twice a day for 60–80 minutes during hospitalization. Outpatients received one-on-one rehabilitation for 20–40 minutes at least once a week.

3. Outcome measures: The three LS risk tests proposed by JOA, the GLFS-25, stand-up test, two-step test, and total CDL stage, were adopted as outcome measures in this before-after comparative study [14]. For evaluation, the CDL and total CDL results of each test were classified into stages 0–3. All measures were performed both before and three months after THA. Data were collected as previously described in Miyazaki et al. [15].

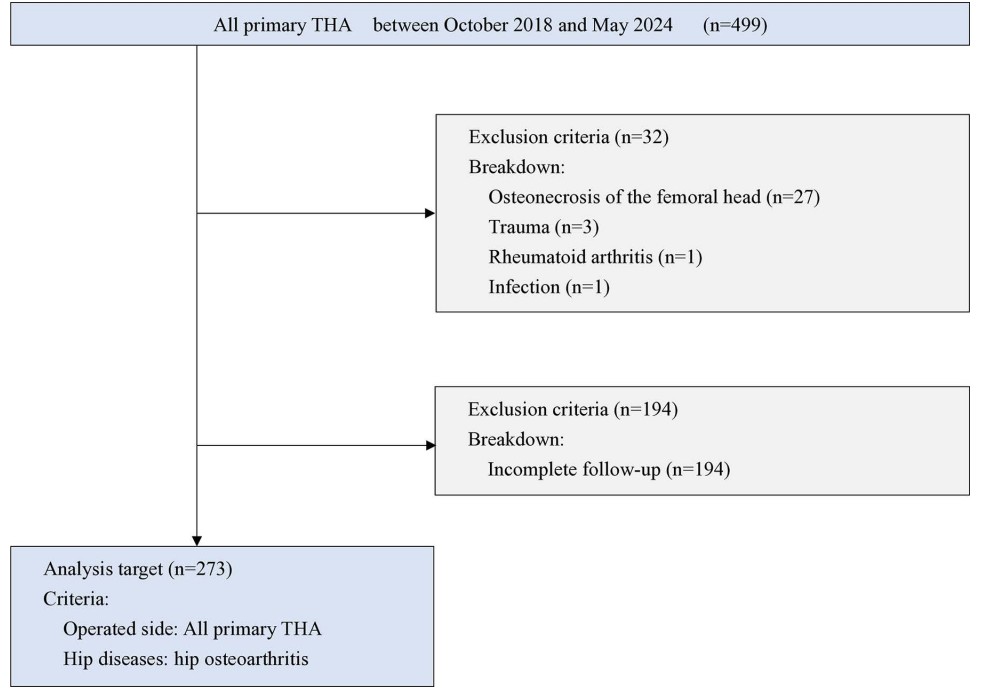

**Fig 1. Flowchart of the included patients.** The participants in this study were patients who underwent primary THA for hip OA between October 2018 and May 2024. Patients with femoral head necrosis, trauma, rheumatoid arthritis, or infection, and patients with incomplete outcome measure data sets were excluded from the study.

The GLFS-25 is a self-administered assessment of 25 questions [11]. This test consists of four questions measuring physical pain, 16 questions measuring ADL, three questions measuring social functions, and two questions measuring mental health in the past month before the test. These 25 questions are rated on a 5-point scale ranging from no functional impairment (0 points) to severe functional impairment (4 points), and a total score is calculated (minimum 0 points, maximum 100 points). Scores ≥24, ≥ 16 to <24, and ≥7 to <16 were classified into CDL stages 3, 2, and 1, respectively.

For the stand-up test, following the JOA guidelines, four different stool heights were used: 40 cm, 30 cm, 20 cm, and 10 cm. First, a patient was asked to stand up from the sitting position starting at the highest stool of 40 cm. After being able to stand up with both legs, the next step was to stand up with one leg. After that, the test continues using stools at different heights. The evaluation employed a 9-point scale [14]: 0 points (unable to stand); 1, 2, 3, and 4 points (stand on both legs from heights of 40 cm, 30 cm, 20 cm, and 10 cm, respectively); 5, 6, 7, and 8 points (stand on one leg from heights of 40 cm, 30 cm, 20 cm, and 10 cm, respectively). Scores <2, <3, and <5 were classified as CDL stages 3, 2, and 1, respectively. The instructions to quantify motion were given. (1) Fold arms in front of the chest at a sitting position. (2) Place feet shoulder-width apart. (3) Position lower legs at an angle of approximately 70° to the floor. (4) Stand up without gaining momentum. (5) Maintain the standing posture for 3 seconds.

The two-step test is used to evaluate stride length. A patient was asked to take two steps with the longest possible stride, and the stride length of these two steps was measured. The test score was calculated by dividing the stride length of these two steps by the patient's height. Scores <0.9, ≥ 0.9 to <1.1, and ≥1.1 to <1.3 were classified into CDL stages 3, 2, and 1, respectively. The instructions to quantify motion were given. (1) Align toes at the starting line with a stationary standing position. (2) Take two of the longest possible steps forward, then align toes together in a stationary standing position. (3) If the subject loses balance, start over. (4) Measure the stride length of the two steps. (5) Perform two times and adopt the better score.

The total CDL was determined based on the results of the stand-up test, two-step test, and GLFS-25. As a result, the most advanced stage of mobility impairment can be determined based on the CDL stages.

## Statistical analysis

The following statistical analyses were performed:

1. Evaluation of LS improvement: The Wilcoxon signed-rank test was used to compare the distribution of the CDL stages for the three LS risk tests and the total CDL stages.

2. Assessment of score changes: Changes in each of the LS test scores and each of the GLFS-25 items before THA and three months after THA were examined using a paired t-test.

3. Comparison of CDL and GLFS-25 improvements: The Chi-squared test was used to examine differences in improvement proportions for each of the GLFS-25 items between the total CDL improvement and non-improvement groups.

This is an observational study with a longitudinal design. At the time of applying for ethical approval, the number of cases over the six-year period could not be predicted. As a result, 273 participants were eligible for this study, all of whom underwent primary THA at the hospital during the study period, excluding those who did not meet the patient selection criteria. All statistical analyses were performed using IBM SPSS 27.0 (IBM Corp., Released 2020. Armonk, NY, USA). The statistical significance was set at $p < .05$.

## Results

Table 1 shows the age, sex, height, weight, and BMI. The participants were 273 patients comprising 52 males and 221 females, with a mean age of 67.4 years (standard deviation SD: 9.3 years). The mean BMI was 24.4 kg/m$^2$ (SD: 3.8 kg/m$^2$). However, this information was excluded from Supporting Information File 1 as it constitutes personal data.

Table 2 shows the participants' medical information. Among the participants in this study, 53 patients (19.4%) had primary hip OA, and 220 (80.6%) patients had secondary hip OA. Regarding THA, 217 (79.5%) patients were unilateral, and 56 (20.5%) patients were bilateral. Regarding the surgical procedure, 156 (57.1%) patients had the AMIS approach, and 117 (42.9%) patients had the transgluteal approach. Regarding the stage on the non-operated side, in the order of stages 0–4, the details were 39 (14.3%), 52 (19.0%), 39 (14.3%), 35 (12.8%), and 45 (16.5%) patients, respectively, and 63 (23.1%) were after THA. On the operated side, in the order of stages 0–4, the details were 0 (0%), 0 (0%), 0 (0%), 33 (12.1%), and 240 (87.9%) patients, respectively.

Table 3 shows the changes observed in total CDL due to THA. THA improved the total CDL in 132 patients with an improvement rate of 48.4%. Before THA, 27 patients were in LS stage 2, of which seven (25.9%) improved to LS stage 1, and one (3.7%) to LS stage 0 after THA. Similarly, among 246 patients who were LS stage 3 before THA, 80 (32.5%) improved to LS stage 2, 43 (17.5%) to LS stage 1, and one (0.4%) to LS stage 0.

**Table 1. Characteristics of the patients.**

| Characteristic | Total (n = 273) |
|---|---|
| Age (years) | 67.4 ± 9.3 |
| Sex: male/female | 52/ 221 |
| Height (cm) | 154.3 ± 7.6 |
| Weight (kg) | 58.4 ± 11.0 |
| Body mass index (BMI) (kg/m$^2$) | 24.4 ± 3.8 |

BMI, body mass index.

Age, height, weight, and BMI values are means ± standard deviation.

**Table 2. Patients' medical information (n=273).**

| Item | Category | Frequency | % |
|---|---|---|---|
| Diagnosis | Primary hip osteoarthritis | 53 | 19.4 |
| | Secondary hip osteoarthritis | 220 | 80.6 |
| THA | Unilateral | 217 | 79.5 |
| | Bilateral | 56 | 20.5 |
| Surgical procedure | AMIS approach | 156 | 57.1 |
| | Transgluteal approach | 117 | 42.9 |
| JOA staging (non-operated side) | Stage 0 | 39 | 14.3 |
| | Stage 1 | 52 | 19.0 |
| | Stage 2 | 39 | 14.3 |
| | Stage 3 | 35 | 12.8 |
| | Stage 4 | 45 | 16.5 |
| | After THA | 63 | 23.1 |
| JOA staging (operated side) | Stage 0 | 0 | 0 |
| | Stage 1 | 0 | 0 |
| | Stage 2 | 0 | 0 |
| | Stage 3 | 33 | 12.1 |
| | Stage 4 | 240 | 87.9 |

THA, total hip arthroplasty; AMIS, anterior minimally invasive surgery; JOA, Japanese Orthopedic Association.

**Table 3. Changes in the total CDL stage by THA.**

| Before THA | 3 months after THA | | | | | | | | | |
|---|---|---|---|---|---|---|---|---|---|---|
| | Stage 0 | | Stage 1 | | Stage 2 | | Stage 3 | | Total | |
| | N | % | N | % | N | % | N | % | N | % |
| Stage 2 | **1** | 3.7 | **7** | 25.9 | 13 | 48.2 | 6 | 22.2 | 27 | 100 |
| Stage 3 | **1** | 0.4 | **43** | 17.5 | **80** | 32.5 | 122 | 49.6 | 246 | 100 |
| Total | 2 | 0.7 | 50 | 18.3 | 93 | 34.1 | 128 | 46.9 | 273 | 100 |

Highlighted numbers indicate improvements (132 patients, 48.4% improvement rate).

CDL, clinical decision limits; THA, total hip arthroplasty.

Table 4 shows the significant differences (p < .001) observed in the scores for all cases in the three LS risk tests (GLFS-25, stand-up test and two-step test) between before THA and three months after THA. The GLFS-25 score was significantly lower at three months after THA. The scores of the stand-up test and the two-step test were significantly higher at three months after THA.

The differences in the GLFS-25 scores before THA and three months after THA are shown using a scatter plot (Fig 2). There is a negative correlation between the GLFS-25 score before THA and the change in the GLFS-25 score (three months after THA minus before THA), and the change in the GLFS-25 score increased in proportion to the GLFS-25 score before THA. As a result of regression analysis, the regression coefficient was −0.723 and the coefficient of determination ($R^2$) was 0.498 (p < .001).

Table 5 shows the average score difference between before and three months after THA for each GLFS-25 item and the percentage of patients who improved in three months after THA compared to the before THA evaluation. For all items, there was a significant difference between the scores before THA and three months after THA; in other words, the scores three months after THA were lower than before THA.

**Table 4. Changes in the three LS test scores and each functional parameter from before to 3 months after THA (n = 273).**

| | Before THA (mean ± SD) | 3 months after THA (mean ± SD) | Paired *t*-test *p*-value |
|---|---|---|---|
| GLFS-25 | 40.9 ± 16.0 | 19.3 ± 12.4 | < 0.001*** |
| Stand-up test | 2.06 ± 1.21 | 2.36 ± 1.17 | < 0.001*** |
| Two-step test | 0.88 ± 0.22 | 1.01 ± 0.20 | < 0.001*** |

LS, locomotive syndrome; THA, total hip arthroplasty; SD, standard deviation;

GLFS-25, 25-Question Geriatric Locomotive Function Scale.

Significantly different:

***$p$ < .001.

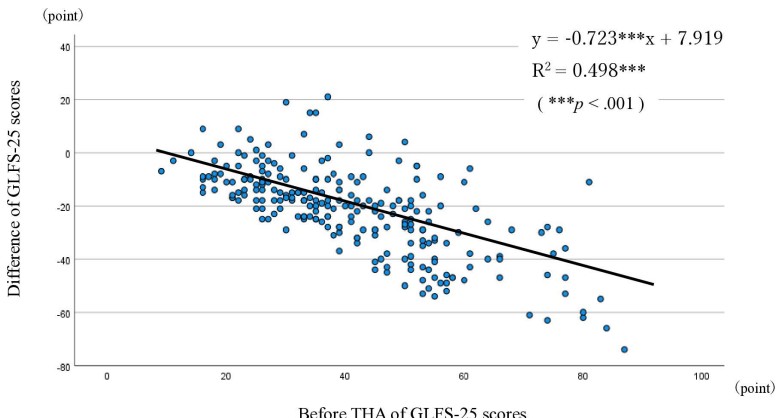

**Fig 2. Scatter plot graph of GLFS-25 score.** The scatter plot shows the correlation between the GLFS-25 score before THA and the change in the GLFS-25 score (three months after THA minus before THA).

Table 6 shows that the percentage of patients who showed improvement in each GLFS-25 item exceeded 60% in 15 out of the 25 items. The highest improvement rate was observed for Q13 (80.2%), followed by Q12 (76.9%), Q3 (76.2%), Q4 (74.7%), Q7 (72.5%), and Q21 (70.7%). In contrast, the lowest improvement rate was observed for Q8 (24.5%). Furthermore, the table shows the number and percentage of patients who showed improvement in each GLFS-25 item, categorized by whether the total CDL stage improved or not. Among patients whose total CDL stage improved, significant.

## Discussion

This study examined LS improvement in patients with hip OA who underwent unilateral primary THA using GLFS-25. Additionally, the impact of LS improvement on each of GLFS-25 item was analyzed. In rehabilitation, a patient's sub-jective symptoms are an important indicator when planning rehabilitation treatment strategies. While previous studies have shown that LS improves after THA, none have clarified how LS improvement affects patients' subjective symptoms. Therefore, this study is the first to focus on GLFS-25, which assesses self-reported musculoskeletal symptoms to clarify its impact on LS improvement. The most significant finding of this study is that improvement in ADL and social function after THA depends on improvement in LS after THA.

Regarding the improvement in GLFS-25 total score, Maezawa et al. [12] reported an improvement from 55.4 points before THA to 19.1 points three months after THA. In this study, GLFS-25 improved from 40.9 points before THA to 19.3 points three months after THA, demonstrating a similar trend to the findings of Maezawa et al. In addition, this study

**Table 5. Changes in GLFS-25 and the improvement rate.**

| Items | Before THA (mean±SD) | 3 months after THA (mean±SD) | Difference (mean±SE) |
|---|---|---|---|
| Q1. Did you have any pain (including numbness) in your neck or upper limbs? | 0.78±0.92 | 0.66±0.82 | 0.12±0.06* |
| Q2. Did you have any pain in your back, lower back or buttocks? | 1.32±1.08 | 0.90±0.92 | 0.42±0.07*** |
| Q3. Did you have any pain (including numbness) in your lower limbs? | 2.23±1.06 | 0.89±0.83 | 1.33±0.08*** |
| Q4. To what extent has it been painful to move your body in daily life? | 2.09±0.92 | 0.82±0.78 | 1.27±0.07*** |
| Q5. To what extent has it been difficult to get up from a bed or lie down? | 1.29±0.99 | 0.43±0.65 | 0.85±0.07*** |
| Q6. To what extent has it been difficult to stand up from a chair? | 1.35±0.99 | 0.44±0.62 | 0.91±0.06*** |
| Q7. To what extent has it been difficult to walk inside the house? | 1.36±0.95 | 0.34±0.58 | 1.03±0.06*** |
| Q8. To what extent has it been difficult to put on and take off shirts? | 0.41±0.77 | 0.14±0.39 | 0.27±0.05*** |
| Q9. To what extent has it been difficult to put on and take off trousers and pants? | 1.80±0.96 | 0.76±0.65 | 1.04±0.06*** |
| Q10. To what extent has it been difficult to use the toilet? | 0.87±0.90 | 0.21±0.44 | 0.66±0.06*** |
| Q11. To what extent has it been difficult to wash your body in the bath? | 1.11±0.97 | 0.47±0.62 | 0.65±0.06*** |
| Q12. To what extent has it been difficult to go up and down stairs? | 2.37±0.99 | 1.06±0.84 | 1.31±0.07*** |
| Q13. To what extent has it been difficult to walk briskly? | 2.62±1.02 | 1.16±0.89 | 1.46±0.07*** |
| Q14. To what extent has it been difficult to keep yourself neat? | 1.01±0.94 | 0.34±0.57 | 0.67±0.06*** |
| Q15. How far can you keep walking without rest? | 2.67±1.12 | 1.52±1.16 | 1.14±0.08*** |
| Q16. To what extent has it been difficult to go out to visit neighbors? | 1.44±1.09 | 0.40±0.66 | 1.04±0.07*** |
| Q17. To what extent has it been difficult to carry objects weighing approximately 2 kilograms? | 1.73±1.23 | 0.82±0.93 | 0.90±0.08*** |
| Q18. To what extent has it been difficult to go out using public transportation? | 2.00±1.26 | 0.86±0.98 | 1.14±0.08*** |
| Q19. To what extent have simple tasks and housework been difficult? | 1.13±0.95 | 0.39±0.63 | 0.74±0.06*** |
| Q20. To what extent have load-bearing tasks and housework been difficult? | 1.81±1.17 | 0.92±0.89 | 0.89±0.07*** |
| Q21. To what extent has it been difficult to perform sports activity? | 2.93±1.03 | 1.70±1.16 | 1.23±0.08*** |
| Q22. Have you been restricted from meeting your friends? | 1.36±1.32 | 1.00±1.18 | 0.36±0.08*** |
| Q23. Have you been restricted from joining social activities? | 2.29±1.46 | 1.66±1.48 | 0.63±0.10*** |
| Q24. Have you ever felt anxious about falls in your house? | 1.30±1.05 | 0.76±0.77 | 0.54±0.07*** |
| Q25. Have you ever felt anxious about being unable to walk in the future? | 1.68±1.16 | 0.64±0.76 | 1.04±0.08*** |

Improvement rate is the comparison of before THA and 3 months after THA.

GLFS-25, 25-Question Geriatric Locomotive Function Scale; THA, total hip arthroplasty; SD, standard deviation; SE, standard error.

Paired *t*-test, Significantly different:

*$p < .05$,

***$p < .001$.

analyzed the changes in each GLFS-25 item. A negative correlation was observed between the GLFS-25 score before THA and the change in the score (three months after THA minus before THA). The decrease in the scores indicates improvement in most patients. Those with higher GLFS-25 scores before THA, indicating greater difficulty in ADL, showed greater improvements after THA. These findings suggest that THA and rehabilitation not only enhance physical function but also have significant impact on patients' subjective symptoms.

The GLFS-25 assesses the following four categories: pain-related (Q1 to Q4), ADL-related (Q5 to Q20), social function-related (Q21 to Q23), and mental health-related (Q24, Q25) [11]. Before THA, items scored as "moderately difficult" (score ≥2) in GLFS-25 included pain-related Q3 and Q4, ADL-related Q12, Q13, Q15, and Q18, and social function-related Q21 and Q23 (Table 5). These results indicate that patients undergoing THA not only experienced hip pain and physical discomfort but also faced significant difficulties in ADL and social activities.

Next, the relationship between the subjective symptom improvements and LS improvements was examined. All items, except pain-related Q3, were significantly associated with improvement in LS. This suggests that LS improvement has

**Table 6. Differences in GLFS-25 improvements based on whether the LS improved or not.**

| items | Total | Total CDL stage | | p |
|---|---|---|---|---|
| | improvement (n = 273) | improvement (n = 132) | non-improvement (n = 141) | |
| Q1 | 84 (30.8) | 43 (32.6) | 41 (29.1) | 0.531 |
| Q2 | 126 (46.2) | 71 (53.8) | 55 (39.0) | 0.014* |
| Q3 | 208 (76.2) | 106 (80.3) | 102 (72.3) | 0.123 |
| Q4 | 204 (74.7) | 111 (84.1) | 93 (66.0) | 0.001** |
| Q5 | 165 (60.4) | 94 (71.2) | 71 (50.4) | 0.000*** |
| Q6 | 165 (60.4) | 94 (71.2) | 71 (50.4) | 0.000*** |
| Q7 | 198 (72.5) | 109 (82.6) | 89 (63.1) | 0.000*** |
| Q8 | 67 (24.5) | 39 (29.5) | 28 (19.9) | 0.063 |
| Q9 | 187 (68.5) | 103 (78.0) | 84 (59.6) | 0.001** |
| Q10 | 138 (50.5) | 78 (59.1) | 60 (42.6) | 0.006** |
| Q11 | 134 (49.1) | 71 (53.8) | 63 (44.7) | 0.133 |
| Q12 | 210 (76.9) | 116 (87.9) | 94 (66.7) | 0.000*** |
| Q13 | 219 (80.2) | 120 (90.9) | 99 (70.2) | 0.000*** |
| Q14 | 139 (50.9) | 82 (62.1) | 57 (40.4) | 0.000*** |
| Q15 | 189 (69.2) | 108 (81.8) | 81 (57.4) | 0.000*** |
| Q16 | 185 (67.8) | 107 (81.1) | 78 (55.3) | 0.000*** |
| Q17 | 166 (60.8) | 94 (71.2) | 72 (51.1) | 0.001** |
| Q18 | 182 (66.7) | 100 (75.8) | 82 (58.2) | 0.002** |
| Q19 | 158 (57.9) | 91 (68.9) | 67 (47.5) | 0.000*** |
| Q20 | 164 (60.1) | 96 (72.7) | 68 (48.2) | 0.000*** |
| Q21 | 193 (70.7) | 109 (82.6) | 84 (59.6) | 0.000*** |
| Q22 | 104 (38.1) | 56 (42.4) | 48 (34.0) | 0.154 |
| Q23 | 131 (48.0) | 76 (57.6) | 55 (39.0) | 0.002** |
| Q24 | 121 (44.3) | 66 (50.0) | 55 (39.0) | 0.068 |
| Q25 | 177 (64.8) | 98 (74.2) | 79 (56.0) | 0.002** |

Values for each GLFS-25 item represent the number and proportion of improved patients.

Chi-square test, Significantly different:

***p < .001,

**p < .01,

*p < .05.

a strong influence on improvements in ADL and social function. One of the greatest advantages of THA is pain relief. As reported by previous studies, THA significantly reduces pain, with its effects lasting over the medium to long term [16–18]. In this study, pain was significantly improved, regardless of the improvement in LS. However, improvement in LS was the requirement for improvement in ADL and social function.

These findings indicate the importance of rehabilitation strategies after THA that focus on enhancing mobility and promoting ADL and social activities. Regarding the relationship between gait characteristics after THA and LS, a previous study reported improvements in overall walking ability and the stance phase (stride length and step length) were associated with improvements in GLFS-25 scores [13]. Based on these findings, focusing on the stance phase during the gait cycle to enhance overall walking ability has been shown to be an effective strategy for improving LS after THA.

Although there are motor functional outcome measures [19,20] to evaluate post-THA rehabilitation effects, there are no indicators that capture changes in subjective symptoms related to daily activities. The strength of this study is that it

focuses on patients' subjective symptoms to evaluate the effects of THA and rehabilitation, providing essential insights for developing rehabilitation strategies. This study has several limitations. First, both unilateral and bilateral hip OA cases were included in the study. Second, both AMIS and transgluteal surgical approaches were used. Third, the cohort included patients with polyarticular disease and those who had undergone arthroplasty in other joints. These factors may have influenced the improvement of GLFS-25. Therefore, future studies are needed to investigate whether these factors affect LS improvement, including outcomes measured by GLFS-25.

## Conclusions

This study examined LS improvement in patients with hip OA who underwent unilateral primary THA, focusing on subjective symptoms using the GLFS-25. Before THA, patients experienced not only severe hip pain and physical discomfort but also significant difficulties with ADL and social activities. Furthermore, LS improvement after THA was strongly associated with improvements in the subjective symptoms of ADL and social functioning. Based on these findings, rehabilitation strategies should focus on supporting patients in enhancing mobility and promoting improvements in ADL and social participation after THA, while optimizing gait function after THA is also crucial for further supporting LS recovery.

## Supporting information

**S1 File. Raw data.**
(XLSX)

**S2 File. Codebook.**
(DOCX)

**S3 File. STROBE Statement.**
(DOCX)

**S4 File. 25-Question Geriatric Locomotive Function Scale.**
(PDF)

## Acknowledgments

The authors would like to thank Masaru Ochiai, Tsubasa Kawaguchi, Aya Unoki, Wakaba Iha, Akari Nagatomo, Kensuke Okamura and Mami Kanno for their assistance in recruiting the participants and for useful advice.

## Author contributions

**Conceptualization:** Shigeaki Miyazaki, Kurumi Tsuruta, Saori Yoshinaga, Yoshinori Fujii, Amy Hombu, Etsuo Chosa.

**Data curation:** Shigeaki Miyazaki, Taro Funamoto, Takero Sakamoto, Takuya Tajima, Yoshihiro Nakamura, Hideki Arakawa, Jun Nakatake.

**Formal analysis:** Kurumi Tsuruta, Yoshinori Fujii.

**Funding acquisition:** Shigeaki Miyazaki.

**Investigation:** Shigeaki Miyazaki, Saori Yoshinaga.

**Methodology:** Shigeaki Miyazaki, Kurumi Tsuruta, Saori Yoshinaga, Yoshinori Fujii, Amy Hombu, Taro Funamoto, Takero Sakamoto, Takuya Tajima, Yoshihiro Nakamura.

**Supervision:** Etsuo Chosa.

**Validation:** Yoshinori Fujii, Taro Funamoto, Takero Sakamoto, Takuya Tajima, Yoshihiro Nakamura.

**Visualization:** Amy Hombu.

**Writing – original draft:** Shigeaki Miyazaki, Kurumi Tsuruta, Saori Yoshinaga, Amy Hombu.

**Writing – review & editing:** Hideki Arakawa, Jun Nakatake, Etsuo Chosa.

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
