## [Decision Letter · Decision Letter 0]

26 Feb 2025

PONE-D-24-52671Investigation of locomotive syndrome improvement by total hip arthroplasty in patients with hip osteoarthritis: a before-after comparative study focusing on 25-question geriatric locomotive function scalePLOS ONE

Dear Dr. Miyazaki,

Thank you for submitting your manuscript to PLOS ONE. After careful consideration, we feel that it has merit but does not fully meet PLOS ONE’s publication criteria as it currently stands. Therefore, we invite you to submit a revised version of the manuscript that addresses the points raised during the review process. Please submit your revised manuscript by Apr 12 2025 11:59PM. If you will need more time than this to complete your revisions, please reply to this message or contact the journal office at plosone@plos.org . Please include the following items when submitting your revised manuscript:

We look forward to receiving your revised manuscript.

Kind regards,

Masaya Anan, Ph.D.

Academic Editor

PLOS ONE

https://www.sciencedirect.com/science/article/abs/pii/S0949265821000385?via%3Dihub

https://pmc.ncbi.nlm.nih.gov/articles/PMC11505882/

In your revision ensure you cite all your sources (including your own works), and quote or rephrase any duplicated text outside the methods section. Further consideration is dependent on these concerns being addressed.

3. We note that there is identifying data in the Supporting Information file < Supporting Information File 1.xlsx>. Due to the inclusion of these potentially identifying data, we have removed this file from your file inventory. Prior to sharing human research participant data, authors should consult with an ethics committee to ensure data are shared in accordance with participant consent and all applicable local laws.

-Location data

Please remove or anonymize all personal information, ensure that the data shared are in accordance with participant consent, and re-upload a fully anonymized data set. Please note that spreadsheet columns with personal information must be removed and not hidden as all hidden columns will appear in the published file.

Reviewers' comments:

Reviewer's Responses to Questions

**Comments to the Author**

1. Is the manuscript technically sound, and do the data support the conclusions?

Reviewer #1: Partly

Reviewer #2: Yes

Reviewer #3: Yes

2. Has the statistical analysis been performed appropriately and rigorously? 

Reviewer #1: I Don't Know

Reviewer #2: Yes

Reviewer #3: Yes

3. Have the authors made all data underlying the findings in their manuscript fully available?

Reviewer #1: Yes

Reviewer #2: Yes

Reviewer #3: Yes

4. Is the manuscript presented in an intelligible fashion and written in standard English?

Reviewer #1: Yes

Reviewer #2: Yes

Reviewer #3: Yes

5. Review Comments to the Author

Reviewer #1: Introduction:

1. Explain the rationale of the study. Please delete information unrelated to objective so that the section is short and sweet. For example, the first page of introduction may be deleted. Kindly focus on three elements of introduction.

a. What is known about the topic? (Background)

b. What is not known? (The research problem)

c. Why the study was done? (Justification)

Methods:

1. Methods section determines the results. Kindly focus on three basic elements of methods section.

a. How the study was designed?

b. How the study was carried out?

c. How the data were analyzed?

Discussion:

1. The discussion section needs to be described scientifically. Kindly frame it along the following lines:

i. Main findings of the present study

ii. Comparison with other studies

iii. Implication and explanation of findings

iv. Strengths and limitations

Reviewer #2: This manuscript presents a valuable contribution to the field of orthopedic medicine, and it offers insights into the short-term outcomes of THA on locomotive syndrome. With minor revisions, it should be suitable for publication in a peer-reviewed journal.

Reviewer #3: General Comments

This paper examines the extent to which LS at 3 months after THA surgery has improved compared to preoperative LS. It acknowledges that the results are robust with a large number of patients. However, while we acknowledge the novelty of comparing LS before and after THA surgery, we feel that the clinical significance and impact is very low.

Specific Comments

BACKGROUND

LS is a concept proposed in Japan that refers to a condition in which locomotion is limited due to musculoskeletal problems. It is partially used outside of Japan, but if it is to have broad clinical significance internationally, it is necessary to compare it with similar concepts such as sarcopenia and frailty and demonstrate the superiority of using LS.

In addition, THA is expected to significantly improve hip function, with associated improvements in locomotion and mobility. In other words, it is a natural surgery for LS to improve. Therefore, it is not clinically meaningful to simply compare LS before and after THA surgery, but we should focus on what items and abilities improve in each evaluation method.

METHODS

I understand that the same is true for the surgical technique of the patient, and that the surgeon is not constant. Regarding rehabilitation, what kind of healthcare professionals are in charge, is there any difference in the content of the rehabilitation, and were all subjects really performed for the duration and frequency described?

Do any of the THA patients include those who had only one side performed and those who had both sides performed at the same time? We expect that LS at 3 months postoperatively will also be strongly affected by contralateral hip OA, and that unilateral and bilateral THA should not be treated in the same way.

RESULTS AND DISCUSSIONS

Some patients have improved LS while others have not, despite having undergone the same surgery. As mentioned earlier, THA is a procedure that has the potential to make a significant contribution to improving physical function, and improvement in LS is a predictable change. We believe that the clinical significance of the data will become clearer when we perform subgroup analysis and discuss the differences between those who improved and those who did not, as well as the differences between the items that improved and those that did not.

6. PLOS authors have the option to publish the peer review history of their article (what does this mean? ). If published, this will include your full peer review and any attached files.

**Do you want your identity to be public for this peer review?** For information about this choice, including consent withdrawal, please see our Privacy Policy .

Reviewer #1: No

Reviewer #2: No

Reviewer #3: No

---

## [Author Response · Author response to Decision Letter 1]

26 Mar 2025

March 27, 2025

Masaya Anan, Academic Editor

PLOS ONE

Dear Dr. Anan:

Thank you for the opportunity to revise our manuscript entitled “Investigation of locomotive syndrome improvement by total hip arthroplasty in patients with hip osteoarthritis: A before-after comparative study focusing on 25-question geriatric locomotive function scale”. We appreciate the reviewers' valuable feedback, which has helped us improve the quality of our manuscript. We believe that the revisions have addressed all comments and enhanced the clarity and rigor of our study.

The revision has been developed in consultation with all coauthors, and each author has given approval to the final form of this revision.

We would greatly appreciate your kind consideration.

Sincerely,

Shigeaki Miyazaki, PhD

Rehabilitation Unit, University of Miyazaki Hospital, Miyazaki, Miyazaki, Japan

Postal address: 5200 Kihara Kiyotake Miyazaki, Miyazaki, 889-1692, Japan

Phone: +81-985-85-9849

E-mail: 03-5-23@med.miyazaki-u.ac.jp

Responses to the comments of Editor

Thank you very much for reviewing our manuscript. We appreciate your time and effort in helping us improve the quality of our work. In response to the comments, we have revised our manuscript and provide detailed responses to each point raised by the reviewers. Our replies listed in the Reply section under each comment.

Editor Comments:

Editor (Masaya Anan)

Thank you for submitting your manuscript to PLOS ONE. After careful consideration, we feel that it has merit but does not fully meet PLOS ONE’s publication criteria as it currently stands. Therefore, we invite you to submit a revised version of the manuscript that addresses the points raised during the review process.

【Reply:】

Thank you for inviting us to revise our manuscript. We have carefully addressed all comments from the reviewers and are pleased to submit the revised version for your consideration. We appreciate your time and review.

Responses to the comments of Reviewers

Thank you very much for reviewing our manuscript. We appreciate your time and effort in helping us improve the quality of our work. In response to the comments, we have revised our manuscript and provide detailed responses to each point raised by the reviewers. Our replies listed in the Reply section under each comment.

Reviewer Comments:

Reviewer 1

Introduction:

1. Explain the rationale of the study. Please delete information unrelated to objective so that the section is short and sweet. For example, the first page of introduction may be deleted. Kindly focus on three elements of introduction.

a. What is known about the topic? (Background)

b. What is not known? (The research problem)

c. Why the study was done? (Justification)

【Reply:】

Thank you for your insightful comments. We have revised the Introduction section by deleting information not directly related to the study objective. We have restructured the section to focus on the three elements as suggested:

a. Background: Lines 89-99

b. The research problem: Lines 99-101

c. Justification: Lines 102-108

Methods:

1. Methods section determines the results. Kindly focus on three basic elements of methods section.

a. How the study was designed?

b. How the study was carried out?

c. How the data were analyzed?

【Reply:】

Thank you for your helpful comments. We have revised the Methods section to address the three basic elements as suggested:

a. Study Design: Lines 111-127

b. Study Execution: Lines 129-198

c. Statistical Analysis: Lines 200-216

Discussion:

1. The discussion section needs to be described scientifically. Kindly frame it along the following lines:

i. Main findings of the present study

ii. Comparison with other studies

iii. Implication and explanation of findings

iv. Strengths and limitations

【Reply:】

Thank you for your valuable comments. We have revised the Discussion section by restructuring it into clearly defined paragraphs as suggested:

i. Main findings of the present study: Lines 279-288

ii. Comparison with other studies: Lines 289-299

iii. Implication and explanation of findings: Lines 300-321

iv. Strengths and limitations: Lines 322-332

Reviewer Comments:

Reviewer 2

This manuscript presents a valuable contribution to the field of orthopedic medicine, and it offers insights into the short-term outcomes of THA on locomotive syndrome. With minor revisions, it should be suitable for publication in a peer-reviewed journal.

Recommendations

1. Include a clear statement on data availability.

2. Consider discussing the limitations of the study, such as the potential impact of polyarticular disease and previous arthroplasties on the results.

3. Elaborate on the clinical implications of the findings, particularly regarding rehabilitation strategies post-THA.

4. Provide more context on how these results compare to existing literature on LS improvement after THA.

【Reply:】

Thank you very much for your encouraging and constructive comments. In response to your recommendations, we have revised the manuscript as follows:

1. Include a clear statement on data availability.

→We have added a Data Availability Statement. (Lines 350-351)

2. Consider discussing the limitations of the study, such as the potential impact of polyarticular disease and previous arthroplasties on the results.

→This point has been added in the sixth paragraph of the Discussion section. (Lines 328-330)

3. Elaborate on the clinical implications of the findings, particularly regarding rehabilitation strategies post-THA.

→We have added the clinical implications of the findings, particularly regarding rehabilitation strategies post-THA in the fifth paragraph of the Discussion section. (Lines 315-321)

4. Provide more context on how these results compare to existing literature on LS improvement after THA.

→We have added additional comparisons with existing literature in the second to fourth paragraphs of the Discussion section. (Lines 289-314)

Reviewer Comments:

Reviewer 3

Specific Comments

BACKGROUND

LS is a concept proposed in Japan that refers to a condition in which locomotion is limited due to musculoskeletal problems. It is partially used outside of Japan, but if it is to have broad clinical significance internationally, it is necessary to compare it with similar concepts such as sarcopenia and frailty and demonstrate the superiority of using LS.

【Reply:】

Thank you for your valuable comments. We have revised the first paragraph of the Introduction to compare LS and similar concepts such as sarcopenia and frailty, and emphasize the advantages of using LS in assessing mobility impairment. (Lines 75-88)

In addition, THA is expected to significantly improve hip function, with associated improvements in locomotion and mobility. In other words, it is a natural surgery for LS to improve. Therefore, it is not clinically meaningful to simply compare LS before and after THA surgery, but we should focus on what items and abilities improve in each evaluation method.

【Reply:】

Thank you for your insightful comments. After thorough discussion among the authors, we fully agreed with your point. Therefore, instead of a simple comparison of total CDL stage before and after THA, we have removed Table 3 and revised the study objectives as follows:

“Although evaluating hip joint function is important, determining which functional abilities are regained is also essential. A comprehensive evaluation using GLFS-25 provides valuable insights into LS improvement after THA and can help refine postoperative rehabilitation strategies. The primary objective of this study is to clarify LS improvement in patients with hip OA who have undergone unilateral primary THA using GLFS-25. A secondary objective is to identify the impact of THA on each specific GLFS-25 item for optimizing functional recovery. “(Lines 102-108)

METHODS

I understand that the same is true for the surgical technique of the patient, and that the surgeon is not constant. Regarding rehabilitation, what kind of healthcare professionals are in charge, is there any difference in the content of the rehabilitation, and were all subjects really performed for the duration and frequency described?

【Reply:】

Thank you for your valuable comments. We have revised the “Surgical approach and post-operative rehabilitation” in the Materials and Methods section to provide additional details that clarify your questions. (Lines 152-161)

Do any of the THA patients include those who had only one side performed and those who had both sides performed at the same time? We expect that LS at 3 months postoperatively will also be strongly affected by contralateral hip OA, and that unilateral and bilateral THA should not be treated in the same way.

【Reply:】

Thank you for your insightful comments. In this study, no patients underwent simultaneous bilateral THA. All cases were unilateral primary THA. We have added this clarification to the “Patient Selection” in Materials and Methods. (Lines 138-139)

RESULTS AND DISCUSSIONS

Some patients have improved LS while others have not, despite having undergone the same surgery. As mentioned earlier, THA is a procedure that has the potential to make a significant contribution to improving physical function, and improvement in LS is a predictable change. We believe that the clinical significance of the data will become clearer when we perform subgroup analysis and discuss the differences between those who improved and those who did not, as well as the differences between the items that improved and those that did not.

【Reply:】

Thank you for your valuable comments. As you suggested, we conducted a subgroup analysis and discussed the differences between items that showed improvement and those that did not. We have added the results to Table 6 (Lines 268-277) in the Results section and to the fourth paragraph of Discussion section. (Lines 307-314).

---

## [Decision Letter · Decision Letter 1]

12 May 2025

Investigation of locomotive syndrome improvement by total hip arthroplasty in patients with hip osteoarthritis: a before-after comparative study focusing on 25-question geriatric locomotive function scale

PONE-D-24-52671R1

Dear Dr. Miyazaki,

We’re pleased to inform you that your manuscript has been judged scientifically suitable for publication and will be formally accepted for publication once it meets all outstanding technical requirements.

Kind regards,

Masaya Anan, Ph.D.

Academic Editor

PLOS ONE

Additional Editor Comments (optional):

Reviewers' comments:

Reviewer's Responses to Questions

**Comments to the Author**

1. If the authors have adequately addressed your comments raised in a previous round of review and you feel that this manuscript is now acceptable for publication, you may indicate that here to bypass the “Comments to the Author” section, enter your conflict of interest statement in the “Confidential to Editor” section, and submit your "Accept" recommendation.

Reviewer #1: All comments have been addressed

Reviewer #3: All comments have been addressed

2. Is the manuscript technically sound, and do the data support the conclusions?

Reviewer #1: Yes

Reviewer #3: Yes

3. Has the statistical analysis been performed appropriately and rigorously? 

Reviewer #1: Yes

Reviewer #3: Yes

4. Have the authors made all data underlying the findings in their manuscript fully available?

Reviewer #1: Yes

Reviewer #3: Yes

5. Is the manuscript presented in an intelligible fashion and written in standard English?

Reviewer #1: Yes

Reviewer #3: Yes

6. Review Comments to the Author

Reviewer #1: THANKS ALOT FOR YOUR REPLAY

THANKS ALOT FOR YOUR REPLAY

THANKS ALOT FOR YOUR REPLAY

THANKS ALOT FOR YOUR REPLAY

Reviewer #3: (No Response)

7. PLOS authors have the option to publish the peer review history of their article (what does this mean? ). If published, this will include your full peer review and any attached files.

**Do you want your identity to be public for this peer review?** For information about this choice, including consent withdrawal, please see our Privacy Policy .

Reviewer #1: No

Reviewer #3: No

---

## [Editor Report · Acceptance letter]

PONE-D-24-52671R1

PLOS ONE

Dear Dr. Miyazaki,

I'm pleased to inform you that your manuscript has been deemed suitable for publication in PLOS ONE. Congratulations! Your manuscript is now being handed over to our production team.

Kind regards,

on behalf of

Dr. Masaya Anan

Academic Editor

PLOS ONE